# Seroprevalence of SARS-CoV-2 and risk factors for infection among children in Uganda: A serial cross-sectional study

Irene Bagala[1,2‡], Jane Frances Namuganga[2‡], Patience Nayebare[2], Gloria Cuu[2], Thomas Katairo[2], Isaiah Nabende[2], Samuel Gonahasa[2], Martha Nassali[2], Stephen Tukwasibwe[2], Grant Dorsey[3], Joaniter I. Nankabirwa[1,2], Sabrina Bakeera-Kitaka[1], Sarah Kiguli[1], Bryan Greenhouse[3], Isaac Ssewanyana[2,4], Moses R. Kamya[1,2], Jessica Briggs[3]*

1 Makerere University College of Health Sciences, Kampala, Uganda, 2 Infectious Diseases Research Collaboration, Kampala, Uganda, 3 University of California, San Francisco, CA, United States of America, 4 Central Public Health Laboratory, Butabika, Uganda

‡ IB and JFN are co-first authors.
* jessica.briggs@ucsf.edu

**Data Availability Statement:** Upon publication, all data will be made available here: https://github.com/jbriggs7/UnMASC_paper.

## Abstract

### Background

Understanding COVID-19's impact on children is vital for public health policy, yet age-specific data is scarce, especially in Uganda. This study examines SARS-CoV-2 seroprevalence and risk factors among Ugandan children at two timepoints, along with COVID-19-related knowledge and practices in households, including adult vaccination status.

### Methods

Baseline surveys were conducted in 12 communities from April to May 2021 (post-Alpha wave) and follow-up surveys in 32 communities from November 2021 to March 2022 (Omicron wave). Household questionnaires and blood samples were collected to test for malaria by microscopy and for SARS-CoV-2 using a Luminex assay. Seroprevalence was estimated at both the survey and community level. Mixed-effects logistic regression models assessed the association between individual and household factors and SARS-CoV-2 seropositivity in children, adjusting for household clustering.

### Results

More households reported disruptions in daily life at baseline compared to follow-up, though economic impacts lingered. By the follow-up survey, 52.7% of adults had received at least one COVID-19 vaccine dose. Overall seroprevalence in children was higher at follow-up compared to baseline (71.6% versus 19.2%, p < 0.001). Seroprevalence in children ranged across communities from 6–37% at baseline and 50–90% at follow-up. At baseline, children from the poorest households were more likely to be infected. Increasing age remained the only consistent risk factor for SARS-CoV-2 seroconversion at both timepoints.

**Funding:** We acknowledge sources of funding support including Makerere University Research and Innovations Fund (MAKRIF/DVCFA/026/20) (IB, JFN, JN, MK, PN, IN); Fogarty International Center of the National Institutes of Health (D43TW010526) (IB, JFN, GC, PN, TK, SG, MN, Dr. ST); the Bill and Melinda Gates Foundation, INV-017893 and INV-023690 (JB, IS, BG); and NIH/NIAID (K23 AI166009) to JB. The funders had no role in study design, data collection and analysis, decision to publish, or preparation of the manuscript.

**Competing interests:** The authors have declared that no competing interests exist.

## Conclusions

Results indicate that a larger number of children were infected during the Delta and Omicron waves of COVID-19 compared to the Alpha wave. This study is the largest seroprevalence survey in children in Uganda, providing evidence that most children were infected with SARS-CoV-2 before the vaccine was widely available to pediatric populations. Pediatric infections were vastly underreported by case counts, highlighting the importance of seroprevalence surveys in assessing disease burden when testing and reporting rates are limited and many cases are mild or asymptomatic.

## Introduction

The emergence of SARS-CoV-2 in 2019 led to a global pandemic in 2020, with countries around the world grappling with its impact. Uganda registered its first case of COVID-19 in March 2020, marking the onset of the Alpha wave [1]. In recognition of the potential for COVID-19 causing significant morbidity and mortality in Uganda, the government responded by instituting lock-down measures including the closure of schools, workplaces, border crossings, and travel restrictions on March 18, 2020 [2].

However, as in many other sub-Saharan African nations, reported cases in Uganda were significantly lower than initially expected [3–5]. As of March 14, 2024, only 172,149 confirmed cases have been reported in Uganda [1], numbers that are likely to be a gross under-estimation of the true extent of the country's COVID-19 disease burden because access to confirmatory testing for SARS-CoV-2 was not widespread and existing surveillance systems were ill-equipped to capture all cases [6]. Notably, there are few age-stratified data available to understand the burden of COVID-19 infection in children, including in Uganda.

Understanding the true burden of COVID-19 infection in pediatric populations is especially challenging because children are more likely to have asymptomatic or mild COVID-19 infection and may therefore be less likely to be captured by surveillance systems that rely on case counts [7–9]. Seroprevalence surveys, which capture both asymptomatic and symptomatic infections, play a vital role in understanding the true burden of SARS-CoV-2 infection [6,10,11]. Evidence from other seroprevalence surveys in Uganda and sub-Saharan Africa suggest high rates of infection despite low national case counts; however, there are no published studies that focus on seroprevalence in children [10–13]. Data from seroprevalence surveys can help policymakers assess the scale of transmission among children and inform public health strategies.

Herein we describe the seroprevalence of SARS-CoV-2 and risk factors for infection among children across Uganda using data from cross-sectional household surveys conducted at two timepoints; April-May 2021 (after the Alpha wave) and November 2021-March 2022 (during the Omicron wave). In addition, these surveys also collected data on COVID-19 related knowledge, attitudes, and practices from an adult member of the household.

## Methods

### Study sites and timeline

This COVID-19 sub-study was embedded within a cluster-randomized trial to evaluate the impact of two types of long-lasting insecticidal bed nets (LLINs) distributed as part of Uganda's 2020–2021 national universal coverage campaign. The parent study included 64 clusters

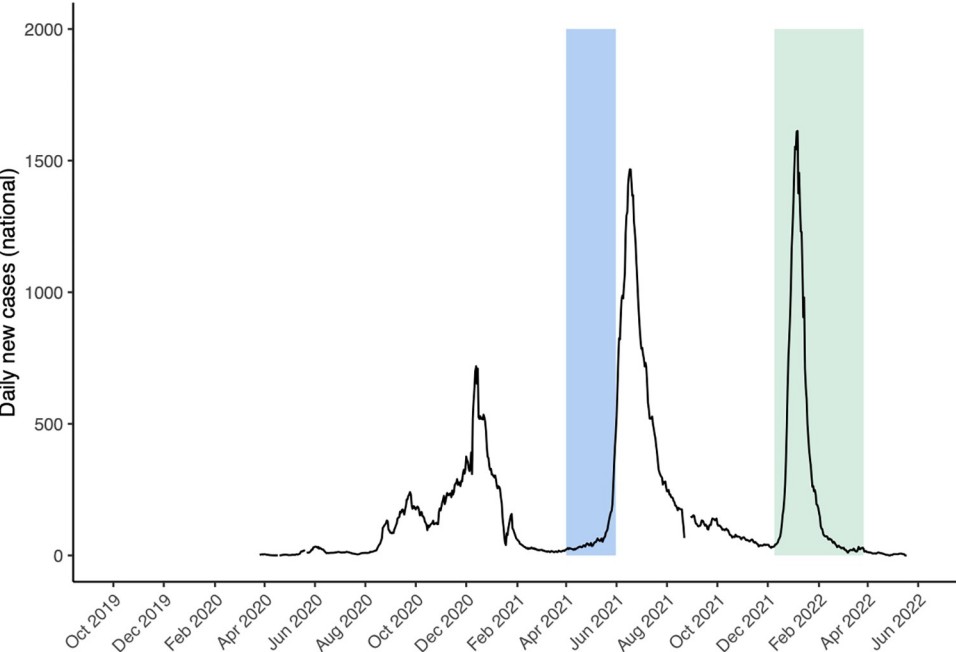

**Fig 1. Timing of the baseline (blue) and follow-up (green) cross-sectional surveys.**

from 32 districts (2 clusters per district). A cluster was defined as a target area surrounding public health facilities with enhanced malaria surveillance, known as Malaria Reference Centres (MRCs). Target areas included the village in which the MRC was located and adjacent villages that met the following criteria: (1) did not contain another government-run health facility, (2) located in the same sub-county as the MRC, and (3) similar incidence of malaria as the MRC's village. All households within the MRC target areas were mapped and enumerated to generate a sampling frame for the cross-sectional surveys. Baseline cross-sectional community surveys were carried out in the target areas of 12 of the 64 MRCs from April to May 2021, and follow-up surveys were carried out in the target areas of all 64 MRCs from November 2021 to March 2022, 12 months following LLIN distribution (**Fig 1**). For each cluster, 50 households with at least one child aged 2–10 were randomly enrolled from those enumerated, using criteria previously published [14].

For the cross-sectional surveys, a household questionnaire was administered to the heads of household or other designated adult using pre-programmed tablets. Information was gathered on characteristics of households and residents, proxy indicators of wealth including ownership of assets, and LLIN ownership. For this sub-study, additional questions on COVID-19 attitudes, beliefs, and behaviors were added. Questions about vaccination status were asked of the household head about every individual in the household.

## Sample collection

During the baseline survey of 12 MRC target areas, a finger prick blood sample was collected from all children present in the household aged 2–10. During the follow-up survey, sample collection was expanded after a protocol modification, and finger prick blood samples were collected from all children less than 18 years of age from 32 of the 64 MRC target areas (one cluster per district). At both timepoints, blood samples were used to prepare a thick blood smear and prepare a dried blood spot (DBS). For both cross-sectional surveys, informed

consent was obtained at the time of the evaluation including consent for future use of biological specimens, along with assent if the child at least 8 years of age. Individual level demographic and clinical data were collected from all children eligible for sample collection.

## Laboratory methods

To determine the presence of malaria parasites, thick blood smears were stained with 2% Giemsa for 30 min and read by experienced laboratory technologists. A thick blood smear was considered negative when the examination of 100 high-power fields did not reveal asexual parasites. For quality control, all slides were read by a second microscopist and discrepant readings were settled by a third reviewer.

For serologic evaluation, serum was eluted from a 6-mm DBS punch using a previously described method [15]. To assess total IgG responses to the receptor-binding domain (RBD) of the spike protein, a Luminex bead assay previously described for SARS-CoV-2 serologic studies [13] was optimized for DBS samples. Samples were assayed at 1:400 dilution to determine antibody seropositivity, with the results expressed as mean fluorescent intensity. A standard curve using a pool of positive serum was included on each plate to normalize for plate-to-plate variations and to infer relative antibody concentrations using a 4-parameter logistic model [16]. For negative controls, 80 DBS were used from the PRISM-2 cohort study in Tororo District, Uganda collected in 2017 and 2018 before the COVID-19 pandemic [17]. For positive controls,151 DBS were used from participants with PCR-confirmed SARS-CoV-2 infection enrolled in the UCSF Long-term Impact of Infection with Novel Coronavirus study [18].

## Statistical analysis

STATA (version 17) and R (version 4.3.1) were used for data analysis. Responses to the household survey about COVID-19 knowledge, attitudes, and behaviors were tabulated for each survey time point as simple proportions and compared using the Chi-squared test. Survey data were included for all 12 communities surveyed at baseline and for the 32 communities with expanded sample collection at follow up.

To determine the cut off for seropositivity, receiver-operator curve analysis was performed using R package ROCR [19] to maximize both sensitivity and specificity of the assay based on the relative antibody concentration data from positive and negative controls. A sample from an individual was determined to be seropositive if the mean fluorescent intensity was above the cutoff. Assay sensitivity was 94.7% and specificity was 97.5%. Raw SARS-CoV-2 seroprevalence at each MRC was calculated as the number of samples that tested positive over all samples tested from that MRC. We then calculated seroprevalence at the MRC level adjusted for sensitivity and specificity.

Univariate and multivariate mixed-effects logistic regression models were used to measure associations between household-level and individual-level factors with SARS-CoV-2 seropositivity among children, accounting for clustering within households. Statistical significance was assessed using two-tailed tests with a p-value threshold of less than 0.05. Confidence intervals for odds ratios were set at 95%.

## Ethical approval

All methods were carried out in accordance with relevant guidelines and regulations including the Declaration of Helsinki. The parent study (LLINEUP2, ClinicalTrials.gov: NCT04566510) was approved by the Institutional Review Boards (IRBs) at Makerere University School of Medicine Research and Ethics Committee (SOMREC) (REF 2020–193), the Uganda National

Council for Science and Technology (UNCST) (HS1097ES), the University of California, San Francisco (20–31769) and London School of Hygiene and Tropical Medicine (22615–1). Written informed consent to participate in the study was obtained from adults or from parents/guardians for their child(ren). A second written consent form was used to consent adults or parents/guardians for the future use of biological specimens obtained during the study. Written assent to participate in the study was also obtained from children aged 8 years and older. A request for waiver of consent to conduct laboratory analysis was obtained from SOMREC. For this sub-study, laboratory analysis was only conducted on samples for which participants had given consent for future use of biological specimens.

## Results

### Study profile

**Fig 1** shows daily reported cases of SARS-CoV-2 at the national level; clear peaks in reported cases are consistent with infections caused by the Alpha variant (December 2020–January 2021), the Delta variant (May–July 2021), and the Omicron variant (December 2021-February 2022) [1]. Data for this study came from cross-sectional surveys performed across Uganda in the target area of 12 MRCs from 10 April to 6 May 2021 (blue bar, **Fig 1**) and the target areas of 32 MRCs from 24 November 2021 to 27 March 2022 (green bar, **Fig 1**). Therefore, the baseline survey data were collected after the Alpha wave and before the surge of the Delta wave, and the follow-up survey data was collected throughout the Omicron wave. Details of the study participants with DBS samples collected at each survey time point are shown in **Fig 2**. In the baseline survey, children ages 2–10 were eligible for sample collection, while in the follow-up survey, all children <18 years of age were eligible for sample collection. Serology data were successfully generated for 96.8% of eligible participants in the baseline study and for 99.4% of eligible participants in the follow-up survey. At the baseline survey, 423 (52.3%) of the children were aged 2–5 years and 386 (47.7%) of the children aged 6–10 years. During the follow-up

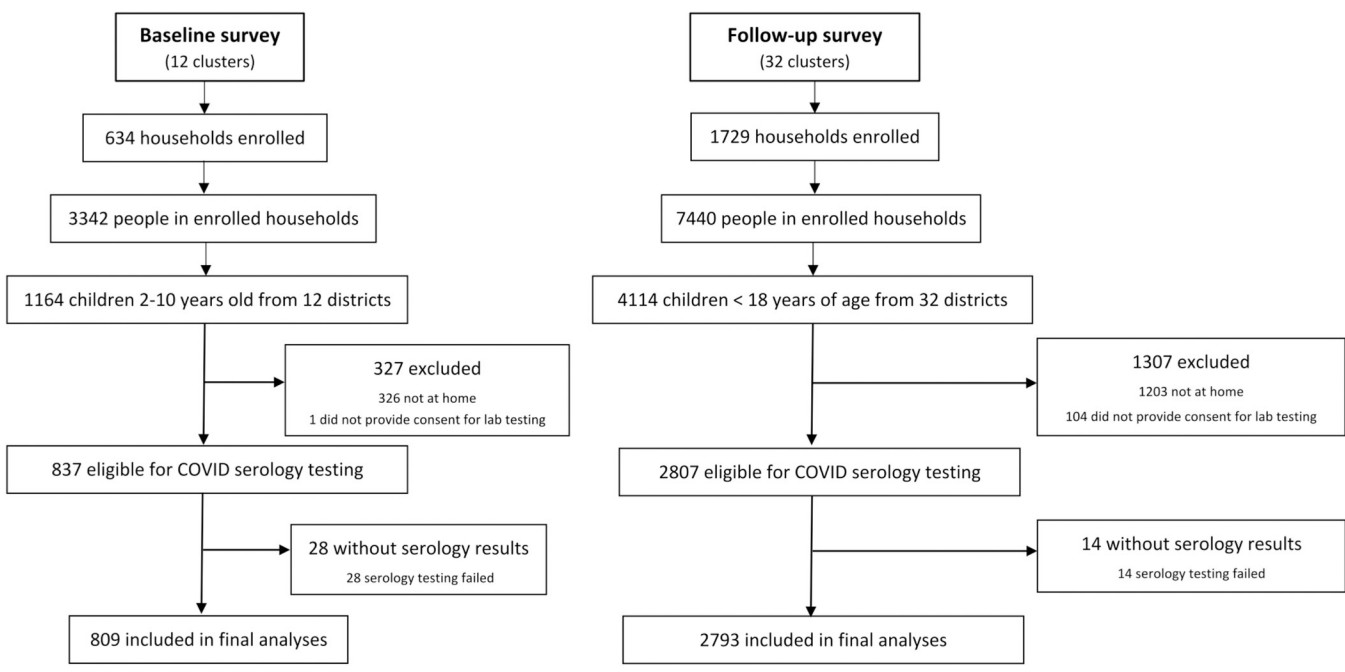

**Fig 2. Study profile of participants enrolled in the study.**

survey, 561 (20.1%) of the children were aged less than 2 years, 973 (34.8%) between 2 to 5 years, 1026 (36.7%) between 6 to 10 years and 233 (8.3%) between 11 to 17 years.

## COVID-19 related knowledge, attitudes, and behaviors

Table 1 presents data on knowledge, attitudes, and behaviors related to COVID-19 from the baseline and the follow-up household surveys. Avoidance of crowds, increased hand washing, and masking outside the home were reported at very high rates (>80%) during the baseline survey and generally remained high, though there was less hand washing reported during the Omicron wave (**Table 1**). Knowledge about the modes of transmission of COVID-19 was also high; > 80% of household heads reported understanding that coughing/sneezing promoted the spread of COVID-19. Face to face talking and indirect spread through fomites were also correctly identified as routes of transmission, though fewer households were aware of fomites as a possible means of transmission. More households reported disruptions in daily life in the baseline survey compared to the follow-up survey (41.3% vs 25.6%). The largest disruptions in daily life at baseline included lost income (21.5%), restricted movement (7.6%), children being unable to attend school (4.6%), and not visiting friends or family (3.2%). All reported disruptions in day-to-day life improved during the follow-up survey; however, 15.8% of households still reported a loss of income, implying that the pandemic had an economic impact still felt at the time of the follow up survey. Compared to the baseline survey, an improvement in ability to travel was reported in the follow-up survey (63.4% vs 49.4% reporting difficulty traveling from baseline to follow-up). Importantly, nearly half of household heads reported health facility medication stockouts at baseline, and 33% reported staff shortages. Both availability of medicines and staff were improved at follow-up but rates of disrupted services remained high.

## COVID-19 vaccination

Vaccination to prevent SARS-CoV-2 infection was not available in Uganda at the time of the baseline survey and was not available for children during the time period of either survey [20].

**Table 1. COVID-19 related knowledge, attitudes, and behaviors.**

| Questions | Categories | Baseline survey (n = 634) | Follow-up survey (n = 1729) |
|---|---|---|---|
| Do you avoid crowds? | | 539 (85.0%) | 1,545 (89.4%) |
| Do you wash your hands more often? | | 551 (86.9%) | 1,280 (74.0%) |
| Do you wear a mask outside the home? | | 553 (87.2%) | 1,419 (82.1%) |
| Can COVID-19 be spread the following way? | Coughing/sneezing | 514 (81.1%) | 1,530 (88.5%) |
| | Face to face talking | 415 (65.5%) | 1,036 (59.9%) |
| | Indirect contact / fomites | 128 (20.2%) | 523 (30.25%) |
| | Doesn't know how COVID-19 is spread | 23 (3.6%) | 19 (1.1%) |
| If your daily life has been changed by COVID-19, how has it changed? | Greater food insecurity | 15 (2.4%) | 23 (1.3%) |
| | Lost income | 136 (21.5%) | 274 (15.8%) |
| | Not visiting friends or family | 20 (3.2%) | 11 (0.6%) |
| | Higher prices | 4 (0.6%) | 16 (0.9%) |
| | Children at home from school | 31 (4.9%) | 31 (1.8%) |
| | Masks required/hand washing/social distancing | 9 (1.4%) | 6 (0.3%) |
| | Movement restricted | 48 (7.6%) | 63 (3.6%) |
| | Lack of medication | 0 (0%) | 0 (0%) |
| | Fear of getting sick | 11 (1.7%) | 28 (1.6%) |
| | Can't go to church | 2 (0.3%) | 12 (0.7%) |
| Travel has been more difficult due to COVID-19 lockdowns | | 402 (63.4%) | 854 (49.4%) |
| Changes noticed regarding care available at local health facilities | No medications available | 303 (47.8%) | 455 (26.3%) |
| | Less staff available | 211 (33.3%) | 221 (12.8%) |

Vaccination status was assessed in follow-up surveys from November 2021 to March 2022 by asking the household head if each individual member of the household was vaccinated. 1,753 individuals out of 3,326 adults (52.7%) were reported to be vaccinated. The majority of individuals received one dose, while 24.6% had received two doses. The most commonly received first vaccine was the Astra-Zeneca vaccine (40.5%), followed by the Johnson and Johnson vaccine (29.5%) and mRNA vaccines (Pfizer, Moderna, or not otherwise specified) (17.4%) (S1 Table). Astra-Zeneca remained the most commonly received second vaccine (59.9%) followed by Sinovac/Sinopharm (18.3%).

## Seroprevalence of SARS-CoV-2 in children at baseline and follow-up surveys

SARS-CoV-2 seroprevalence in children was estimated in the target areas around each MRC for both survey timepoints (Fig 3). Overall raw and adjusted seroprevalence was higher at follow-up compared to baseline (raw seroprevalence 73.5% versus 19.4%; adjusted seroprevalence 71.6% versus 19.2%, p < 0.001). In the baseline survey, seroprevalence in the communities surrounding the surveyed MRCs ranged from 5.7% at Butagaya to 37.3% at Namokora. Seroprevalence increased at all sites in the follow-up survey; the lowest seroprevalence at follow-up was 50.0% at Kigandalo, and 9 sites had a seroprevalence >81% in the follow-up survey, with the highest seroprevalence at Kibaale (89.6%).

## Individual and household level risk factors associated with seropositivity to SARS-COV-2 among Ugandan children

Table 2 presents associations between individual and household risk factors and SARS-CoV-2 seroprevalence among children in Uganda by univariate and multivariate analysis. At baseline, older age was associated with seroconversion (aOR 1.69, CI 1.11–2.57). In addition, children in the poorest households were more likely to have seroconverted compared to those in the wealthiest households (aOR 2.51, CI 1.37–4.60). At follow-up, only increasing age was associated with SARS-CoV-2 seroconversion; there was no longer an association between household wealth index and seroconversion.

Gender was not associated with seroconversion at baseline or at follow up. Microscopic malaria parasitemia had no association with SARS-CoV-2 seroconversion at either survey

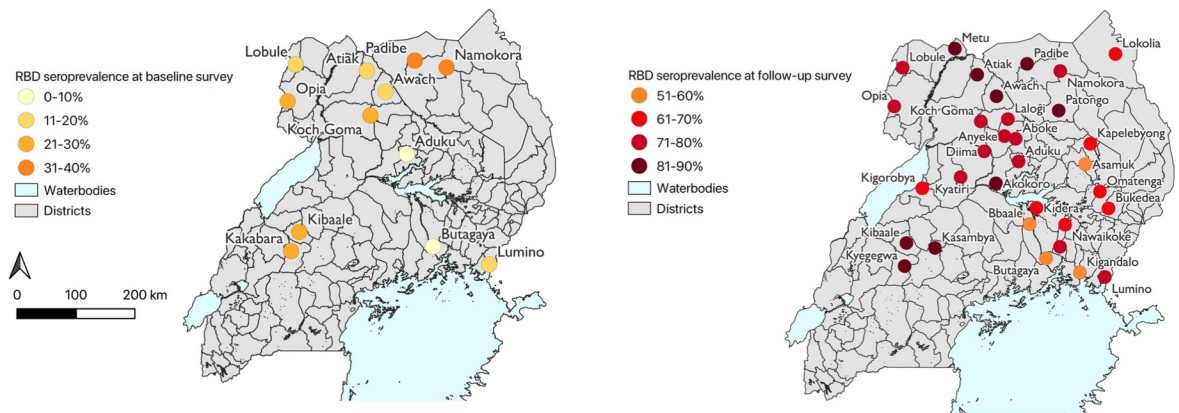

**Fig 3. SARS-CoV-2 seroprevalence in the target areas surrounding MRCs in the baseline and follow-up cross-sectional surveys. A.** RBD seroprevalence in children ages 2–10 in the baseline survey. **B.** RBD seroprevalence in children 0–17 in the follow-up survey. Source for Uganda district shapefiles: https://data.humdata.org/dataset/uganda-administrative-boundaries-as-of-17-08-2018.

**Table 2. Risk factors associated with seropositivity to SARS-COV-2 among Ugandan children at baseline and follow-up surveys.**

| Variable | Categories | Baseline survey | | | | | Follow-up survey | | | | |
|---|---|---|---|---|---|---|---|---|---|---|---|
| | | Seroprevalence | Univariate analysis* | | Multivariate analysis* | | Seroprevalence | Univariate analysis* | | Multivariate analysis* | |
| | | | OR (95% CI) | p-value | aOR (95% CI) | p-value | | OR (95% CI) | p-value | aOR (95% CI) | p-value |
| **Individual level risk factors** | | | | | | | | | | | |
| Age in years | ≤ 2 | – | – | | – | | 371/561 (66.1%) | reference group | | reference group | |
| | 3–5 | 68/423 (16.1%) | reference group | | reference group | | 695/973 (71.4%) | 1.39 (1.05–1.85) | 0.02 | 1.42 (1.07–1.90) | 0.015 |
| | 6–10 | 89/386 (23.1%) | 1.72 (1.13–2.64) | 0.01 | 1.69 (1.11–2.57) | 0.02 | 797/1026 (77.7%) | 2.03 (1.52–2.72) | 0.00 | 2.11 (1.56–2.84) | 0.000 |
| | 11–17 | – | – | | – | | 190/233 (81.6%) | 3.13 (1.95–5.04) | 0.00 | 3.16 (1.96–5.09) | 0.000 |
| Gender | Male | 72/401 (18.0%) | reference group | | reference group | | 973/1340 (72.6%) | reference group | | reference group | |
| | Female | 85/408 (20.8%) | 1.21 (0.81–1.81) | 0.35 | 1.16 (0.78–1.74) | 0.46 | 1080/1453 (74.3%) | 1.16 (0.94–1.44) | 0.16 | 1.11 (0.89–1.38) | 0.34 |
| Parasitemia by microscopy | No | 115/588 (19.6%) | reference group | | reference group | | 1579/2151 (73.4%) | reference group | | reference group | |
| | Yes | 42/220 (19.1%) | 0.82 (0.60–1.5) | 0.82 | 0.83 (0.52–1.32) | 0.43 | 474/642 (73.8%) | 1.04 (0.80–1.34) | 0.79 | 0.89 (0.68–1.17) | 0.42 |
| **Household level risk factors** | | | | | | | | | | | |
| Wealth index | Least poor | 36/260 (13.9%) | reference group | | reference group | | 772/1037 (74.5%) | reference group | | reference group | |
| | Middle | 54/282 (19.2%) | 1.51 (0.89–2.56) | 0.125 | 1.55 (0.88–2.72) | 0.128 | 652/896 (72.8%) | 0.96 (0.72–1.29) | 0.79 | 0.89 (0.65–1.22) | 0.48 |
| | Poorest | 67/265 (25.3%) | 2.27 (1.34–3.84) | 0.002 | 2.51 (1.37–4.60) | 0.003 | 629/860 (73.1%) | 0.98 (0.73–1.31) | 0.88 | 0.82 (0.57–1.17) | 0.27 |
| House construction | Modern | 44/215 (20.5%) | reference group | | reference group | | 831/1144 (72.6%) | reference group | | reference group | |
| | Traditional | 113/594 (19.0%) | 0.88 (0.55–1.42) | 0.62 | 0.61 (0.36–1.06) | 0.08 | 1222/1649 (74.1%) | 1.10 (0.86–1.41) | 0.44 | 1.00 (0.71–1.41) | 0.99 |
| Number of residents per room used for sleeping | <2 | 30/185 (16.2%) | reference group | | reference group | | 390/527 (74.0%) | reference group | | reference group | |
| | 2–3 | 77/399 (19.3%) | 1.29 (0.75–2.23) | 0.36 | 1.11 (0.64–1.94) | 0.71 | 1097/1498 (73.2%) | 0.96 (0.70–1.31) | 0.79 | 1.00 (0.72–1.53) | 0.99 |
| | >3 | 50/225 (22.2%) | 1.58 (0.86–2.90) | 0.14 | 1.32 (0.71–2.46) | 0.39 | 566/768 (73.7%) | 1.00 (0.70–1.44) | 1.00 | 1.05 (0.71–1.53) | 0.82 |
| Presence of windows | Uncovered windows | 14/62 (22.6%) | reference group | | reference group | | 59/92 (64.1%) | reference group | | reference group | |
| | Covered windows | 62/360 (17.2%) | 0.67 (0.30–1.49) | 0.32 | 62/360 (17.2%) | 0.48 | 1000/1380 (72.5%) | 1.50 (0.76–2.97) | 0.25 | 1.38 (0.69–2.77) | 0.36 |
| | No windows present | 81/387 (20.9%) | 0.88 (0.40–1.93) | 0.74 | 81/387 (20.9%) | 0.58 | 994/1321 (75.3%) | 1.88 (0.95–4.62) | 0.07 | 1.97 (0.96–4.02) | 0.64 |

timepoint. At the household level, there was no association between house construction type, household crowding as measured by the number of residents per room for sleeping, or type of windows and SARS-CoV-2 seroprevalence at baseline or follow up.

## Discussion

Results from the study indicate a drastic increase in the SARS-CoV-2 seroprevalence in children from the baseline survey to the follow up survey, indicating that a larger proportion of children were infected during the Delta and Omicron waves of COVID-19 compared to the Alpha wave. Overall seroprevalence at the follow-up survey was 72%, showing that the majority of children had been infected by SARS-CoV-2 by early 2022. Older age was associated with

increasing SARS-CoV-2 seroprevalence both at baseline and during the follow-up survey. Interestingly, in the baseline survey children from the poorest households were more likely to have been infected by SARS-CoV-2 compared to children from the wealthiest households, but this association was no longer evident later in the epidemic after additional waves of infection. In addition, disruptions to daily life and access to medications and healthcare were more significant earlier in the epidemic but improved over time, with the exception of lingering economic impacts such as lost income. COVID-19 vaccination had reached over half of surveyed adults by the time of the follow-up survey.

While there are no other published studies from Uganda on SARS-CoV-2 seroprevalence exclusively in children, our finding of an overall seroprevalence of 72% in a survey conducted from November 2021-March 2022 (during the Omicron wave) is consistent with seroprevalence studies conducted in Uganda in other age groups. A blood donor study that evaluated donors aged 16 and over between October 2019 and April 2022 found that N and S seropositivity increased throughout the pandemic, reaching 83% in January-April 2022 [12]. A cohort study in eastern Uganda that included both children and adults found that by the end of the Delta wave (before widespread vaccination) 68% of the cohort had been infected, and after the Omicron wave, 85% had been infected [13]. While the seroprevalence estimate of 72% from this study is lower than that found by the two other studies in Uganda after the Omicron wave, this might be expected since this study was conducted during and not after the Omicron wave. In addition, since increasing age is correlated with increasing seroprevalence, a lower seroprevalence may be expected in a study that only includes children compared to studies including adults.

The finding that increasing age is associated with higher seroprevalence in children is in agreement with seroprevalence studies in Uganda and elsewhere that have indicated seroprevalence is lowest in young children [8,11–13,21–23]. Higher seroprevalence with increasing age could be due to differences in behavior, with older children more likely to attend gatherings outside the household [24,25], immunity [26–28], or case severity [7,29,30]. We also found evidence that children from poorer households were more likely to be infected with SARS-CoV-2 earlier in the pandemic. This is consistent with studies from other countries where lower socioeconomic status was associated with a higher risk of infection early in the pandemic [21,25,31]. However, by the time of the follow-up survey, this association was no longer evident, likely because the majority of children had already been infected and we did not have the ability to test for re-infection. Therefore, while it is possible that the poorest children remained at higher risk and were re-infected at higher rates than children in wealthier households, we were unable to test this hypothesis.

Survey findings indicated severe disruptions in daily life, including inability to travel and lack of access to medications at local public health facilities. Other significant reported disruptions included loss of income and food insecurity. These disruptions improved over time but had not normalized to the pre-COVID era by the follow-up survey. This is consistent with findings in a previous study in Uganda, Ghana and other low income countries that reported negative effects on finances and food insecurity as well as limited access to medical services during the COVID-19 pandemic [32–35]. Previous studies have also reported disruptions in medical service provision including stockouts of drugs [35] and disruptions in health service utilization [36]. One modeling study using district-level DHIS2 data from Uganda reported negative effects of COVID-19 on utilization of health services, including outpatient attendance at public health facilities and child health services, which were most severe early in the pandemic [37]. This is in contrast with another study by Namuganga et al. showing no major effects on the total number of visits to 17 sentinel malaria surveillance outpatient health facilities in the first year after the COVID-19 pandemic [2]. Most studies that do report disruptions

due to the COVID-19 pandemic report the highest effect in the earliest months of the pandemic, consistent with the trend seen in this study and with the timing of the strictest lockdown policies in Uganda.

Our study is the largest seroprevalence survey in children in Uganda, providing evidence that most children were infected with SARS-CoV-2 before the vaccine was available to pediatric populations. Though only 12 districts were included in the baseline survey, data was available from 32 districts in the follow-up survey, providing good geographic representation of most regions in Uganda except for the southwest. A limitation is that SARS-CoV-2 seroprevalence may have been underestimated in this study because SARS-CoV-2 antibodies wane over time, particularly in asymptomatic or mild infection [38,39]. In addition, most study sites were rural, and therefore some more urban areas such as Wakiso District and Kampala that registered higher numbers of SARS-CoV-2 cases (due to higher transmission or better access to testing) during pandemic waves were not represented in this survey. This would also likely bias our seroprevalence estimate toward underestimation. Our study highlights the usefulness of seroprevalence surveys to estimate the true burden of infection, especially in rural populations with less access to molecular testing for SARS-CoV-2 where using case reports alone would severely underestimate the burden of SARS-CoV-2 infection in children.

## Conclusions

In summary, SARS-CoV-2 seroprevalence among children in Uganda was high by early 2022, providing evidence that the majority of children in Uganda were infected before the availability of pediatric vaccination. Assessment of risk factors associated with SARS-CoV-2 infection revealed increasing risk with increasing age, and higher risk in poorer children. These findings may be useful to direct public health preventative strategies to protect vulnerable pediatric populations in the next global pandemic. In addition, because COVID-19 disease in children is often asymptomatic or mild and molecular testing for infection was not easily accessible, case reports of infection in children vastly underrepresented the true infection rate in children in Uganda. Seroprevalence surveys remain an important way of assessing the true burden of disease when testing and reporting rates are not readily available.

## Supporting information

**S1 Table. Vaccination status in 3,326 adults at follow-up survey.**
(DOCX)

## Acknowledgments

We would like to thank the LLINEUP2 study team and all of the participants who participated in the study.

## Author Contributions

**Conceptualization:** Bryan Greenhouse, Isaac Ssewanyana, Jessica Briggs.

**Data curation:** Patience Nayebare, Thomas Katairo, Isaiah Nabende, Jessica Briggs.

**Formal analysis:** Irene Bagala, Jane Frances Namuganga, Patience Nayebare, Thomas Katairo, Jessica Briggs.

**Funding acquisition:** Joaniter I. Nankabirwa, Jessica Briggs.

**Investigation:** Patience Nayebare, Gloria Cuu, Jessica Briggs.

**Methodology:** Patience Nayebare, Samuel Gonahasa, Jessica Briggs.

**Project administration:** Jane Frances Namuganga, Isaiah Nabende, Samuel Gonahasa, Martha Nassali.

**Resources:** Grant Dorsey, Isaac Ssewanyana, Moses R. Kamya.

**Software:** Isaiah Nabende.

**Supervision:** Jane Frances Namuganga, Samuel Gonahasa, Martha Nassali, Stephen Tukwasibwe, Joaniter I. Nankabirwa, Sabrina Bakeera-Kitaka, Sarah Kiguli, Isaac Ssewanyana, Moses R. Kamya, Jessica Briggs.

**Validation:** Jessica Briggs.

**Visualization:** Jessica Briggs.

**Writing – original draft:** Irene Bagala, Jane Frances Namuganga, Jessica Briggs.

**Writing – review & editing:** Irene Bagala, Patience Nayebare, Grant Dorsey, Joaniter I. Nankabirwa, Sabrina Bakeera-Kitaka, Sarah Kiguli, Bryan Greenhouse, Moses R. Kamya, Jessica Briggs.

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
