## [Decision Letter · Decision Letter 0]

8 Aug 2024

PONE-D-24-24474Seroprevalence of SARS-CoV-2 and risk factors for infection among children in Uganda: a serial cross-sectional studyPLOS ONE

Dear Dr. Briggs,

I would like to sincerely apologise for the delay you have incurred with your submission. It has been exceptionally difficult to secure reviewers to evaluate your study. We have now received three completed reviews; the comments are available below. One of the reviewers feels more positive about your work but the other reviewers have raised significant scientific concerns about the study that need to be addressed in a revision.

Please revise the manuscript to address all the reviewer's comments in a point-by-point response in order to ensure it is meeting the journal's publication criteria. Please note that the revised manuscript will need to undergo further review, we thus cannot at this point anticipate the outcome of the evaluation process.

We look forward to receiving your revised manuscript.

Kind regards,

Miquel Vall-llosera Camps

Senior Staff Editor

PLOS ONE

Journal Requirements:

   "We acknowledge sources of funding support including Makerere University Research and Innovations Fund (MAKRIF/DVCFA/026/20) (IB, JFN, JN, MK, PN, IN);  Fogarty International Center of the National Institutes of Health (D43TW010526) (IB, JFN, GC, PN, TK, SG, MN, Dr. ST); the Bill and Melinda Gates Foundation, INV-017893 and INV-023690 (JB, IS, BG); and NIH/NIAID (K23 AI166009) to JB. "

6. We note that Figure 3 in your submission contain map/satellite images which may be copyrighted. All PLOS content is published under the Creative Commons Attribution License (CC BY 4.0), which means that the manuscript, images, and Supporting Information files will be freely available online, and any third party is permitted to access, download, copy, distribute, and use these materials in any way, even commercially, with proper attribution. For these reasons, we cannot publish previously copyrighted maps or satellite images created using proprietary data, such as Google software (Google Maps, Street View, and Earth). For more information, see our copyright guidelines: http://journals.plos.org/plosone/s/licenses-and-copyright.

a. You may seek permission from the original copyright holder of Figure 3 to publish the content specifically under the CC BY 4.0 license.  

Reviewers' comments:

Reviewer's Responses to Questions

**Comments to the Author**

1. Is the manuscript technically sound, and do the data support the conclusions?

Reviewer #1: Yes

Reviewer #2: Yes

Reviewer #3: Yes

2. Has the statistical analysis been performed appropriately and rigorously? 

Reviewer #1: Yes

Reviewer #2: Yes

Reviewer #3: Yes

3. Have the authors made all data underlying the findings in their manuscript fully available?

Reviewer #1: Yes

Reviewer #2: No

Reviewer #3: Yes

4. Is the manuscript presented in an intelligible fashion and written in standard English?

Reviewer #1: Yes

Reviewer #2: Yes

Reviewer #3: Yes

5. Review Comments to the Author

Reviewer #1: This is a manuscript on an important subject of COVID 19 and prevalence risk factors and effect on vulnerable population.It is well written but there are several clarifications that needs to be done to make it even better.

-Please clarify who is the Corresponding author is. is it Jessica Briggs or Irene Bagala?

-Since you have not reported testing The SARS COV2 strains, you can only conclude that the seroprevalence --was 'during' the said strain was circulating and not caused 'by' the said strains

-Not clear how the 12 and the 32 communities were picked out of 64 MRC

-Not clear why the consent to use stored samples was waived yet you had access to the population ( what reasons did you give the ETHICS review board?)

-I feel the word “Impact” is too strong, could you replace with another word such as “effect”

-You introduce the use of samples from other studies without giving a short brief on the studies.

-At baseline you do not have results for under 2 years- any reason?

-Please state if an In-house determination of cut of using R package ROCR would affect comparability with other studies?.

Reviewer #2: Bagala et al. describe the results of a cross-sectional survey nested in a cluster-randomized trial conducted in Uganda. This report informs children’s exposure to SARS-CoV-2 across epidemic waves while analyzing knowledge, attitudes, and behavior toward COVID-19 at the household level. Overall, the manuscript is well-written and presents relevant data. However, the following points need to be clarified/improved:

Background

L56. “SARS-CoV-2 acronym already includes virus in it. So, there might be no need to add “virus” after SARS-CoV-2.

Methods

L99-100. “For each cluster for each survey…” Could the authors rephrase to avoid repeating “each”?

L111. Authors might need to rephrase as follows for a better understanding of the sentence: “… prick blood sample was collected from all children aged 2-10 present in the household.”

L116-117. “…along with assent if the child was 8 years old or older…” could be replaced with “…along with assent if the child was at least eight years of age or at least eight years old”.

L160-164. The authors state that four institutions approved the study. Yet, they didn’t provide approval numbers. “LLINEUP2” seems to be the project name, and other details provided in L160 refer to the clinical trial registration of the main study.

L190. Please add the content in parenthesis rather after “study” than after “The parent.”

Results

A general concern is that the authors mention in the abstract and methods that they adjusted the seroprevalence to account for clustering within households and for the test kit performance. Could they provide crude seroprevalence estimates to see whether adjustments significantly changed the figures?

L175, 179, 180. I wonder why Figure 2 appears before Figure 1 while Figure 1 appears for the first time in the Results section.

L188. “…the children were ages…” This is likely a typo, please correct.

L192 vs L193. Are the authors presenting data on attitudes or perceptions? Both seem to be used interchangeably in the next paragraphs.

L175-226. Although data on knowledge, attitudes, behaviors, and vaccination are important to understanding COVID-19, they seem to draw the reader’s attention more than the actual seroprevalence data, which is supposed to be the focus of this study. Presenting seroprevalence data first to address the main research question and then mentioning additional data could be one way forward.

L295-297. Please rephrase the sentence for a better understanding.

Figure 3. Please indicate the north and the scales on the provided maps of RBD seroprevalence at baseline and follow-up

L373-377. Please provide the approval number for the listed ethical approvals

Reviewer #3: This is a straightforward analysis of a substudy conducted within the context of a large, cluster-randomized trial. The statistical analysis is appropriate and well-conducted, although I would like to see an enumeration of the variables included in the multivariate models. The manuscript is clearly written, with cogent discussions of study limitations. I have little to recommend in the way of changes. I might quibble over whether households' omitting fomites as a SARS-CoV-2 transmission truly represents a lack of knowledge (I'm not sure fomites have ever been shown to be even a moderate exposure risk), but this is minor and debatable. The results have modest immediate utility, but assessing "importance" is beyond the scope of review. In any case, as the largest seroprevalence survey of children in Uganda to date, the study is valuable.

6. PLOS authors have the option to publish the peer review history of their article (what does this mean?). If published, this will include your full peer review and any attached files.

Reviewer #1: **Yes: **Dr Evans Amukoye

Reviewer #2: No

Reviewer #3: No

---

## [Author Response · Author response to Decision Letter 0]

18 Sep 2024

I attached the response to reviewers as a word document, but will copy below as well. 

Reviewer #1: This is a manuscript on an important subject of COVID 19 and prevalence risk factors and effect on vulnerable population. It is well written but there are several clarifications that needs to be done to make it even better.

-Please clarify who is the Corresponding author is. Is it Jessica Briggs or Irene Bagala?

I have updated this to reflect that it is Jessica Briggs after discussing with Irene.

-Since you have not reported testing The SARS COV2 strains, you can only conclude that the seroprevalence --was 'during' the said strain was circulating and not caused 'by' the said strains

-Changed the word "by" to "during" in the abstract (line 47) and in the Discussion, (line 280). 

-Not clear how the 12 and the 32 communities were picked out of 64 MRC

Funding was only available to test samples from 12 of the communities at baseline. Before the follow-up survey, additional funding was obtained and the protocol was amended to allow us to survey all ages in 32 of the 64 communities instead of children 2-10 only. There were 2 MRCs per district, so 1 MRC from each district was chosen for the all-ages survey. Only samples from children aged 0-17 were tested in the follow up survey as detailed in the methods. Edited line 120 in the methods to reflect the protocol modification and line 122 to explain that one cluster was chosen per district. 

-Not clear why the consent to use stored samples was waived yet you had access to the population ( what reasons did you give the ETHICS review board?)

During the initial collection of samples for the LLINEUP2 cross-sectional surveys, participants could consent to "future use of biological samples." Therefore, we were only allowed to test samples from those patients who had consented to future use of the dried blood spots collected. Even so, we had to obtain an additional waiver from SOMREC to perform this COVID-19 sub-study using these de-identified samples, as detailed in lines 180-181.

-I feel the word “Impact” is too strong, could you replace with another word such as “effect”

I am unsure which use of the word "impact" the reviewer is referring to. If it is the use of it in the abstract, I would argue that "effect" is a synonym for "impact" and many papers that we cited (see references) even use the term impact in their titles. (eg References 2, 32, and 34). 

-You introduce the use of samples from other studies without giving a short brief on the studies.

Lines 142-143 have been edited. The negative controls were obtained from a cohort study conducted before the COVID-19 pandemic, and the positive controls were obtained from participants with PCR-confirmed SARS-CoV-2 infection in a study at UCSF. Also, these studies are referenced, so that the reader can find the original studies if interested. 

-At baseline you do not have results for under 2 years- any reason?

This was a sub-study conceived after the LLINEUP2 randomized controlled trial had started; the original version of the trial protocol did not allow for collection of samples in children <2 years. We were able to modify the protocol to collect samples from children <18 years in time for the follow-up survey. 

-Please state if an In-house determination of cut of using R package ROCR would affect comparability with other studies?

Comparability to other studies would be affected not by the use of ROCR which uses standard formulas to calculate sensitivity and specificity, but by the characteristics of the Luminex assay and the positive and negative controls used. Therefore, we compared our results to other published SARS-CoV-2 serologic data in Uganda (lines 296-301) and found that our results were consistent with seroprevalence studies conducted in Uganda in other age groups. 

Reviewer #2: Bagala et al. describe the results of a cross-sectional survey nested in a cluster-randomized trial conducted in Uganda. This report informs children’s exposure to SARS-CoV-2 across epidemic waves while analyzing knowledge, attitudes, and behavior toward COVID-19 at the household level. Overall, the manuscript is well-written and presents relevant data. However, the following points need to be clarified/improved:

Background

L56. “SARS-CoV-2 acronym already includes virus in it. So, there might be no need to add “virus” after SARS-CoV-2.

Fixed on line 58. 

Methods

L99-100. “For each cluster for each survey…” Could the authors rephrase to avoid repeating “each”?

Fixed, line 104-105. 

L111. Authors might need to rephrase as follows for a better understanding of the sentence: “… prick blood sample was collected from all children aged 2-10 present in the household.”

Fixed lines 115-116 

L116-117. “…along with assent if the child was 8 years old or older…” could be replaced with “…along with assent if the child was at least eight years of age or at least eight years old”.

Done, lines 124-125. 

L160-164. The authors state that four institutions approved the study. Yet, they didn’t provide approval numbers. “LLINEUP2” seems to be the project name, and other details provided in L160 refer to the clinical trial registration of the main study.

L190. Please add the content in parenthesis rather after “study” than after “The parent.”

Fixed; lines 171-176, and also in the ethics statement in the declarations. 

Results

A general concern is that the authors mention in the abstract and methods that they adjusted the seroprevalence to account for clustering within households and for the test kit performance. Could they provide crude seroprevalence estimates to see whether adjustments significantly changed the figures?

Raw seroprevalence at each survey timepoint can be calculated from Table 2, but I have also added these figures into the text at lines 249-251. As you can see, the adjustment for characteristics of the assay results in minor changes to the raw seroprevalence figures. The effect of household clustering was adjusted for only in the logistic regression models used to generate the results in Table 2. 

L175, 179, 180. I wonder why Figure 2 appears before Figure 1 while Figure 1 appears for the first time in the Results section.

Fixed, and figures re-labeled correctly. 

L188. “…the children were ages…” This is likely a typo, please correct.

Fixed, line 205

L192 vs L193. Are the authors presenting data on attitudes or perceptions? Both seem to be used interchangeably in the next paragraphs.

The word "perception" was only used once; it has been changed to "attitude" on line 210. 

L175-226. Although data on knowledge, attitudes, behaviors, and vaccination are important to understanding COVID-19, they seem to draw the reader’s attention more than the actual seroprevalence data, which is supposed to be the focus of this study. Presenting seroprevalence data first to address the main research question and then mentioning additional data could be one way forward.

Changing the order of the results per this reviewer's opinion would require restructuring of the paper. Because this was a concern brought up by only one of three reviewers, and not by any of the co-authors upon review of the paper, we would prefer to keep the results as they are presented currently.

L295-297. Please rephrase the sentence for a better understanding.

Fixed, lines 315-319. 

Figure 3. Please indicate the north and the scales on the provided maps of RBD seroprevalence at baseline and follow-up

Added north and scales on new Figure 3. 

L373-377. Please provide the approval number for the listed ethical approvals

Done as stated above. 

Reviewer #3: This is a straightforward analysis of a sub-study conducted within the context of a large, cluster-randomized trial. The statistical analysis is appropriate and well-conducted, although I would like to see an enumeration of the variables included in the multivariate models. The manuscript is clearly written, with cogent discussions of study limitations. I have little to recommend in the way of changes. I might quibble over whether households' omitting fomites as a SARS-CoV-2 transmission truly represents a lack of knowledge (I'm not sure fomites have ever been shown to be even a moderate exposure risk), but this is minor and debatable. The results have modest immediate utility, but assessing "importance" is beyond the scope of review. In any case, as the largest seroprevalence survey of children in Uganda to date, the study is valuable.

Variables included in the multivariate models can be found in Table 2.

---

## [Decision Letter · Decision Letter 1]

9 Oct 2024

Seroprevalence of SARS-CoV-2 and risk factors for infection among children in Uganda: a serial cross-sectional study

PONE-D-24-24474R1

Dear Dr. Briggs,

We’re pleased to inform you that your manuscript has been judged scientifically suitable for publication and will be formally accepted for publication once it meets all outstanding technical requirements.

Within one week, you’ll receive an e-mail detailing the required amendments. I would especially raise your attention on the declaration of publicly making the data available used in the manuscript. This declaration is currently missing in your submission. When these have been addressed, you’ll receive a formal acceptance letter and your manuscript will be scheduled for publication.

Kind regards,

Sk Md Mamunur Rahman Malik

Academic Editor

PLOS ONE

Additional Editor Comments (optional):

Reviewers' comments:

Reviewer's Responses to Questions

**Comments to the Author**

1. If the authors have adequately addressed your comments raised in a previous round of review and you feel that this manuscript is now acceptable for publication, you may indicate that here to bypass the “Comments to the Author” section, enter your conflict of interest statement in the “Confidential to Editor” section, and submit your "Accept" recommendation.

Reviewer #1: All comments have been addressed

Reviewer #2: All comments have been addressed

Reviewer #3: All comments have been addressed

2. Is the manuscript technically sound, and do the data support the conclusions?

Reviewer #1: Yes

Reviewer #2: Yes

Reviewer #3: (No Response)

3. Has the statistical analysis been performed appropriately and rigorously? 

Reviewer #1: Yes

Reviewer #2: Yes

Reviewer #3: (No Response)

4. Have the authors made all data underlying the findings in their manuscript fully available?

Reviewer #1: Yes

Reviewer #2: (No Response)

Reviewer #3: (No Response)

5. Is the manuscript presented in an intelligible fashion and written in standard English?

Reviewer #1: Yes

Reviewer #2: (No Response)

Reviewer #3: (No Response)

6. Review Comments to the Author

Reviewer #1: You have addressed all my concerns except use of the term 'Impact' which to me means long term effect. Ill treat it as a matter of semantics

Reviewer #2: All comments have been addressed although line numbers with changes were hard to find.

Authors might need to check the following:

L141. “specimens, along with assent if the child “was” at least 8 years of age. “was” is missing.

Reviewer #3: (No Response)

7. PLOS authors have the option to publish the peer review history of their article (what does this mean?). If published, this will include your full peer review and any attached files.

Reviewer #1: No

Reviewer #2: No

Reviewer #3: No

---

## [Editor Report · Acceptance letter]

15 Oct 2024

PONE-D-24-24474R1 

PLOS ONE

Dear Dr. Briggs, 

I'm pleased to inform you that your manuscript has been deemed suitable for publication in PLOS ONE. Congratulations! Your manuscript is now being handed over to our production team.

Kind regards, 

on behalf of

Dr. Sk Md Mamunur Rahman Malik 

Academic Editor

PLOS ONE